# Colour Duplex and/or Contrast-Enhanced Ultrasound Compared with Computed Tomography Angiography for Endoleak Detection after Endovascular Abdominal Aortic Aneurysm Repair: A Systematic Review and Meta-Analysis

**DOI:** 10.3390/jcm11133628

**Published:** 2022-06-23

**Authors:** Georgios I. Karaolanis, Constantine N. Antonopoulos, Efstratios Georgakarakos, Georgios D. Lianos, Michail Mitsis, Georgios K. Glantzounis, Athanasios Giannoukas, George Kouvelos

**Affiliations:** 1Vascular Unit, Department of Surgery, University Hospital of Ioannina and School of Medicine, 455 00 Ioannina, Greece; 2Department of Vascular Surgery, “Attikon” University Hospital, National and Kapodistrian University of Athens, 124 61 Athens, Greece; kostas.antonopoulos@gmail.com; 3Department of Vascular Surgery, University Hospital of Alexandroupolis, “Democritus” University of Thrace, 681 00 Alexandroupolis, Greece; efstratiosgeorg@gmail.com; 4Department of Surgery, School of Medicine, University of Ioannina, 451 10 Ioannina, Greece; georgiolianos@yahoo.gr (G.D.L.); mmitsis@uoi.gr (M.M.); gglantzounis@uoi.gr (G.K.G.); 5Department of Vascular Surgery, Faculty of Medicine, School of Health Sciences, University Hospital of Larissa, University of Thessaly, 411 10 Larissa, Greece; giannouk@uth.gr (A.G.); geokouv@gmail.com (G.K.)

**Keywords:** endovascular aortic repair (EVAR), abdominal aortic aneurysm, endoleaks, Duplex Ultrasound, Contrast-enhanced ultrasound

## Abstract

This study aims to assess the role of Color Duplex Ultrasound with or without contrast media for surveillance following endovascular aortic aneurysm repair (EVAR). A systematic search of the literature published until April 2022 was performed according to the Preferred Reporting Items for Systematic Reviews and Meta-Analyses (PRISMA) guidelines. The pooled rates of endoleak detection through Contrast-Enhanced or Color Duplex Ultrasound (CEUS or CDUS) and Computed Tomography Angiography (CTA) with 95% confidence intervals (CIs) were estimated using random-effect analysis. Thirty-eight studies were considered eligible for inclusion. The total number of patients in the included studies was 5214 between 1997 and 2021. The overall pooled rate of endoleak detection using CDUS and CTA was 82.59% and 97.22%, while the rates for CEUS and CTA were 96.67% and 92.82%, respectively. The findings of the present study support the use of the CEUS for endoleak detection. However, it should be integrated into institutional protocols for EVAR surveillance to further evaluate its clinical utility in the post-EVAR period before it can be recommended as the sole imaging modality after EVAR.

## 1. Introduction

An increasing number of patients suffering from abdominal aortic aneurysm (AAA) are undergoing endovascular aortic repair (EVAR) as opposed to open surgery. The benefits of EVAR have been proved over time regarding both early and midterm postoperative morbidity and mortality [1,2,3,4]. However, complications, such as endoleaks can pose a significant threat to these patients, leading to more interventions [5]. Endoleak incidence varies by type, and ranges from 4% to 10% for type I and III endoleaks, and from 10% to 27% for type II endoleaks [5]. Lifelong surveillance is necessary to detect an endoleak early and avoid the consequent threat of rupture [6,7,8].

Computed Tomography Angiography (CTA) has been the most common modality adopted for surveillance [7,8]. However, it has significant drawbacks such as contrast induced nephropathy, stochastic risk of radiation-induced cancer and cost [7,8]. In order to minimize these events, both the European Society of Vascular Surgery (ESVS) and the North American Society of Vascular Surgery (SVS) recommend the use of colour duplex ultrasound (CDUS) as an accurate imaging tool for postoperative surveillance [7,8]. The adjunction of contrast media to the CDUS (Contrast-Enhanced Ultrasound/CEUS) increased the ability to detect endoleaks and direct re-intervention, evidence that has been confirmed in a number of studies [9,10]. However, studies comparing CDUS with CEUS in detecting endoleaks are sparse in the literature [11].

The objective of the present study was to examine the diagnostic accuracy in terms of sensitivity, specificity, negative and positive predictive value of CDUS and CEUS compared with CTA for endoleak detection after EVAR. A meta-analysis providing pooled rates of endoleak detection for the three modalities was also performed.

## 2. Material and Methods

### 2.1. Information Sources

The Preferred Reporting Items for Systematic Reviews and Meta-Analyses (PRISMA) guidelines were used for this meta-analysis [12]. We systematically searched Medline (database provider PubMed, from 1966 to April 2022), Scopus, EMBASE (database provider Ovid, from 1980 to April 2022), and Cochrane Central Register of Controlled Trials (April 2022) for articles of interest. We also performed a snowball process in the reference lists of the eligible articles to capture additional eligible articles. We applied the snowball process as a technique to reveal further relevant and potentially eligible studies by tracking the citations of all eligible studies.

### 2.2. Search Methodology

We used the following search terms in all possible combinations: “((abdominal aortic aneurysm) OR (endovascular aortic repair)), AND ((duplex ultrasound) OR (contrast-enhanced ultrasound)) AND (surveillance)”. All English-language scientific papers published up to April 2022 were potentially eligible. Two authors (G.K., C.N.A.) independently extracted and analyzed the data and the final decision was reached by consensus. The Newcastle–Ottawa tool (NOS) was applied to evaluate the methodologic quality of the studies [13]. This scale was developed to assess the quality of studies using a “star system” (maximum nine stars), in which a study is judged on three broad perspectives: (1) the selection of the study groups, (2) the comparability of the groups, and (3) the ascertainment of outcome of interest [13].

### 2.3. Inclusion and Exclusion Criteria

All the comparison studies on CDUS and/or CEUS and CTA providing data on the detection of endoleak following EVAR were included in the present meta-analysis. Studies focusing only on one imaging technique, studies providing mixed results or studies that did not provide numerical data were excluded from the analysis. We also excluded editorial, commentary and review articles. Duplicates were excluded, and in case of metachronous publications from the same surgical group, only the latest article or the article with the greatest number of patients was included.

### 2.4. Data Extraction

Data extracted from eligible studies included the first author’s name, study year, study design, total number of patients who underwent surveillance at regular intervals during follow-up by both CDUS (with or without contrast) and CTA scan, number of operators performing CDUS and/or CEUS, type of aortic stents and the mean follow-up (months). The primary outcome was defined as the number of patients detected with an endoleak by CDUS and/or CEUS and CTA scan. Secondary outcomes included the number and type of endoleaks detected during surveillance, the sensitivity, specificity, accuracy, positive and negative predictive value of both CDUS (with or without contrast) and CTA scan.

Endoleaks were defined the persistence of blood flow within the aneurysm sac but outside the lumen of the aortic endograft [7]. A categorization of endoleaks was proposed as follows: type I (inadequate seal at proximal or distal end of the endograft): type II (patent aortic branch vessels such as lumbar, inferior mesenteric artery, accessory renal and hypogastric arteries) that demonstrated collateral filling and back bleeding into the aneurysm sac; type III (disconnection of module of fabric disruption); type IV (porosity of the aortic endograft); and type V (continued increase of the aneurysm sac without demonstrable signs on imaging) [7].

### 2.5. Statistical Analyses

We extracted the number of patients with endoleaks from each of the eligible studies and we thereafter reported them as the proportion of patients with endoleaks among all patients for studies comparing CDUS vs. CTA and CEUS vs. CTA. All values of the studied outcomes were subsequently expressed as proportions and 95% confidence intervals (95% CIs) and thereafter transformed into quantities according to the Freeman–Tukey variant of the arcsine square root transformed proportion. The pooled effect estimates were calculated as the back-transformation of the weighted mean of the transformed proportions using the DerSimonian–Laird weights of the random effects model and expressed as percent proportions. We used a formal statistical test for heterogeneity (I^2^ test). Publication bias was assessed using the Egger’s test for small study effects, as well as visual inspection of funnel plots. The STATA statistical software v14 (StataCorp LP, College Station, TX, USA) was used for our analyses.

## 3. Results

### 3.1. Identification of Relevant Studies

A total of 1540 study titles were identified by the initial search strategy. A review of the titles and abstracts identified that 1435 articles were irrelevant at the first screening stage. One hundred five manuscripts were further evaluated. Of the eligible publications, 78 were excluded for one or more of the following reasons: studies providing data with miscellaneous results (*n* = 2); no comparative studies (*n* = 7); studies which did not provide results for the outcomes of interest (*n* = 5); review articles (*n* = 44); irrelevant studies (*n* = 13); commentary (*n* = 2); editorial (*n* = 1); non-English (*n* = 3); studies with duplicated data/patients (*n* = 1).

Finally, after applying these exclusions, 38 studies [5,9,10,11,14,15,16,17,18,19,20,21,22,23,24,25,26,27,28,29,30,31,32,33,34,35,36,37,38,39,40,41,42,43,44,45] corresponding to a total of 5214 patients were identified as potentially eligible for inclusion in the systematic review and meta-analysis after the addition of 11 studies [9,11,16,21,27,31,32,33,34,36,38] resulting from the snowball process (Figure 1). Twenty-seven studies [5,14,15,16,17,18,19,20,21,22,23,24,25,26,27,28,29,30,31,32,33,34,35,36,37,38,39] with an overall number of 3583 participants provided data comparing the CDUS with CTA, while 15 studies [9,10,11,19,23,26,28,29,39,40,41,42,43,44,45] with a total number of 1631 patients reported comparative data rely on CEUS and CTA imaging modalities. Six studies [19,23,26,28,29,39] from the same center were included in the present meta-analysis because they provided data separately either for CDUS or CEUS with CTA. Moreover, we included in the present meta-analysis the first published study from the same surgical group instead of the metachronous series, due to presence of more data regarding the main outcome. All the eligible studies of the present meta-analysis are illustrated in Table 1 and Table 2.

Baseline study characteristics of the 43 eligible studies included in the systematic review are presented in Table 1 and Table 2. The included studies were published from 1997 to 2021. Eight studies [14,15,16,18,20,22,25,34] with a total number of 1196 patients comparing CDUS with CTA reported data regarding the comorbidities of the participants. Of these, 635 (53%) were heavy smokers, 1101 (92%) suffered from hypertension, 273 (23%) had diabetes melitus, 527 (44%) had chronic obstructive pulmonary disease, 763 (64%) had dyslipidemia, 255 (21%) had chronic renal failure, 948 (79%) had coronary artery diseases and 122 (10%) had peripheral arterial diseases. A total of 122 (10%) and 420 (35%) patients had American Society of Anaesthesiologists (ASA) scores of I-II and III-IV, respectively.

Thirteen studies [15,16,17,18,22,23,25,26,30,32,34,37,38] (2482 patients) reported the brand names of the stents. The most used type of stent was Zenith (29%), followed by AneuRx (19%), Excluder (17%), Talent (10%), Endurant (5%) and other commercially aortic endografts (20%).

Twenty-three studies [5,14,17,18,19,20,21,22,23,24,25,26,27,28,29,30,31,33,34,35,36,38,39] (3583 patients) compared CDUS and CTA and revealed 887 and 1049 endoleaks, respectively. Using CDUS, type I endoleaks were detected in 24% of patients (*n* = 211), type II in 72% (*n* = 645), type III in 3% (*n* = 26), type IV (*n* = 1) and type V (*n* = 4) < 1%. Using CTA, type I endoleaks appeared in 25% of patients (*n* = 262), type II in 71% (*n* = 746), type III in 3% (*n* = 26), type IV < 1% (*n* = 2) and type V (*n* = 13) in 1%.

Fifteen studies [9,10,11,19,23,26,28,29,39,40,41,42,43,44,45] (1631 patients) presented data regarding the type of endoleak detected by CEUS and CTA. Using CEUS, type I, II and III endoleaks were detected in 13% (*n* = 76), 84% (*n* = 481), and 3% (*n* = 16) of patients, respectively. Neither type IV nor type V endoleaks were reported. Using CTA, type I, II and III endoleaks were detected in 12% (*n* = 62), 83% (*n* = 444) and 4% (*n* = 22) of patients, respectively. Type IV and V were also reported in <1%.

In most of the studies, CDUS and CEUS were performed by a single experienced physician [11,19,20,21,22,25,27,30,37,40,41,42,43], whereas in other studies two [14,23,26,31,36,38,44] and three or more physicians were responsible for the scans [9,10,16,18,29,34].

### 3.2. Sensitivity-Specificity-Accuracy and Negative or Positive Predictive Value of CDUS and CEUS

Twenty studies [14,15,16,17,18,19,21,22,23,24,25,27,28,30,31,32,34,35,37,38,39] reported data regarding the sensitivity, specificity, positive (PPV) and negative predictive value (NPV) of CDUS to detect endoleak compared with CTA. The sensitivity of CDUS ranged from 40% to 100% [19,32]. The specificity ranged from 84% to 100% [14,24]. The NPV ranged from 71% to 100% [32,38] and the PPV ranged from 39% to 100% [14,15,38]. The accuracy of the method ranged from 82% to 99% [32,38] (Table 1).

Eleven studies reported data regarding the properties of CEUS compared with CTA for the detection of endoleaks. The sensitivity ranged from 44% to 100% [19,26,42]. The specificity ranged from 82% to 100% [11,19,23,40,44]. The NPV ranged from 44% to 100% [42,45], and the PPV ranged from 91% to 100% [44,45]. The accuracy of the method ranged from 89% to 99% [11,23] (Table 2).

### 3.3. Meta-Analysis

#### 3.3.1. Rates of Endoleak Detection with CDUS and CTA

In all the eligible studies [5,14,15,17,18,19,20,21,22,23,24,25,26,27,28,29,30,31,32,33,34,35,36,37,39], the pooled rate of endoleak detection was 82.59% (95% CI: 69.01–93.23) (Figure 2) and 97.22% (95% CI: 93.13–99.73) (Figure 3) for CDUS and CTA, respectively.

#### 3.3.2. Rates of Endoleak Detection with CEUS and CTA

In all the eligible studies [9,10,11,19,23,26,28,29,39,40,41,42,43,44] the pooled rate of endoleak detection was 96.67% (95% CI: 88.72–100) (Figure 4) and 92.82% (95% CI: 77.39–100) (Figure 5) for CEUS and CTA, respectively.

## 4. Discussion

This meta-analysis derived from a comprehensive review of retrospective studies provides the most contemporary pooled endoleak outcome rates detected by CDUS and/or CEUS and CTA for patients undergoing EVAR. The pooled rates of endoleak detection were 82.59% and 97.22% for CDUS and CTA, respectively, and when comparing CEUS with CTA, the pooled rates were 96.67% and 92.82%, respectively.

Surveillance after EVAR is universally accepted even though there is currently no ideal frequency or standard regimen. The aim of this surveillance is to predict or detect complications, such as endoleaks or migration of the main graft, postoperatively. Both the ESVS (class I, level of evidence B) and the SVS (Level of recommendation 1 (strong), Quality of evidence B (Moderate)) currently recommend CTA scanning at 1 and 12 months during the first year after EVAR, and if neither endoleak nor aneurysm expansion is detected subsequently, a CDUS follow-up may be a reasonable alternative [7,8].

Although CTA has been characterized as the gold standard imaging modality for the assessment and detection of most EVAR complications due to its ability to perform up to three scans (native, arterial, and delayed phase contrast imaging), some negative aspects (contrast-induced nephropathy, ionising radiation, high cost) limit its frequently repeated use [8]. On the other side, CDUS, which is readily available and non-invasive, offers the possibility of repeated and reliable measurement of maximum aneurysm diameter to detect endoleaks [8].

One decade ago, Mirtza et al. [46], comparing CDUS with CT in 21 studies, reported a pooled rate of endoleak detection by CDUS of 77% (95% CI: 0.64–0.86). Several years later, one more meta-analysis presented similar findings of 74% (95% CI: 0.62–0.83) sensitivity [47]. In our study, the pooled rate of endoleak detection was 82.59%; however, this outcome is derived from 27 studies with a large number of participants and imaging performance.

The addition of microbubbles as ultrasound contrast (CEUS), seems to increase the sensitivity of this imaging modality for endoleak detection [8]. Despite the lack of evidence on the use of its imaging modality, a number of studies in the literature have reported a high accuracy in comparison with CTA [11,26,29]. A systematic review published in 2010 considered seven eligible studies with 288 patients [46] and revealed that the sensitivity of CEUS was 98% (95% CI: 0.90–0.99) for the detection of endoleaks after EVAR compared with CTA. Several years later, another study considered eight studies with an overall number of 454 patients and established that the pooled sensitivity of CEUS for endoleak detection was 91% [48]. In both studies, the authors suggested that CEUS demonstrates a highly sensitive modality for endoleak detection in comparison with CTA, especially in delayed endoleaks of type II. In the present study, a similar trend was observed with the CEUS pooled rate of endoleak detection being slightly higher at 96.67% (95% CI: 88.72–100) than that for CTA.

The downsides of CDUS and CEUS are their dependence on the operator and their level of experience, patient related factors (e.g., obesity, hernias, heavily calcified vessels) and the inability to assess the sealing zone length, stent-graft overlap and device migration. In the former case, the present meta-analysis showed that in most of the cases, only one physician performed both ultrasound modalities in the same session in the same sequence every time. As a result, the risk of intra-observer error is not stratified, although we accept this has probably limited the risk of inter-observer error. It is worth noting that using a second or even third operator would have been of great benefit for settling external validity. Therefore, the minimum number of supervised physicians required for EVAR surveillance using both ultrasound modalities remains an unresolved topic.

There are several limitations of this study, mainly mirroring the limitations of the included studies. Firstly, the surveillance protocol after EVAR is very heterogeneous, with surveillance protocols based on different imaging modalities, frequency of imaging and length of follow-up. Secondly, unlike CTA, the reliability of CDUS and CEUS is accompanied by operator dependency, and its practice requires experience. Ultrasound surveillance of EVAR treated patients requires experienced sonographers but would also draw attention to the fact that little research has been undertaken to identify and overcome the challenges associated with the implementation of vascular ultrasound.

## 5. Conclusions

CDUS is an imaging modality commonly used with CTA in post-EVAR follow-up. Our study highlighted that CEUS may offer a safe and sensitive modality for endoleak detection. However, it should be integrated into institutional protocols for EVAR surveillance, potentially obviating the need for patient exposure to high radiation doses and nephrotoxic agents in recurrent CTA scans. Further studies with a larger number of patients and experienced physicians are required to evaluate the clinical safety of CEUS and its utility in the post-EVAR period before it can be recommended as the sole imaging modality after EVAR.

## Figures and Tables

**Figure 1 jcm-11-03628-f001:**
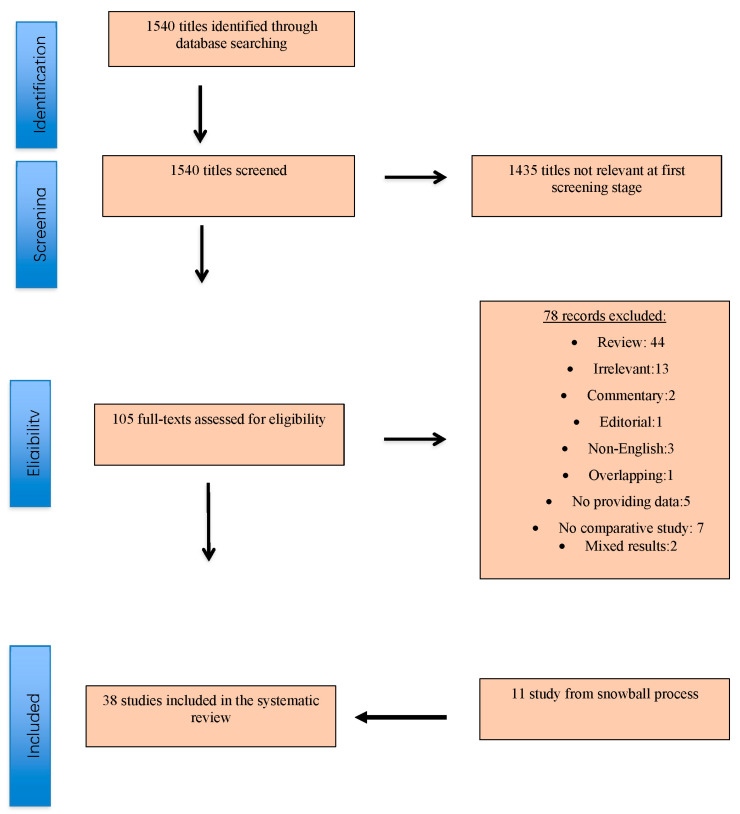
Study flow chart (“Preferred Reporting Items for Systematic reviews and Meta-Analysis” diagram).

**Figure 2 jcm-11-03628-f002:**
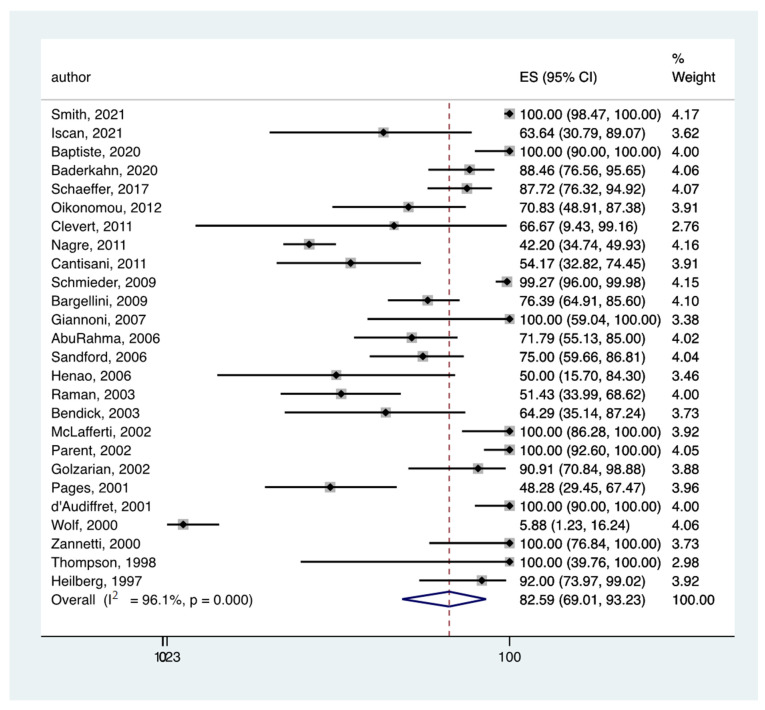
Forest plot presenting the meta-analysis the pooled rate of endoleak detection for CDUS. Event rates in the individual studies presented as squares, with 95% confidence interval (CIs) presented as extending lines. The pooled event rate with its 95% CI is depicted as a diamond. ES: Effect Estimate [5,14,15,17,18,19,20,21,22,23,24,25,26,27,28,29,30,31,32,33,34,35,36,37,39].

**Figure 3 jcm-11-03628-f003:**
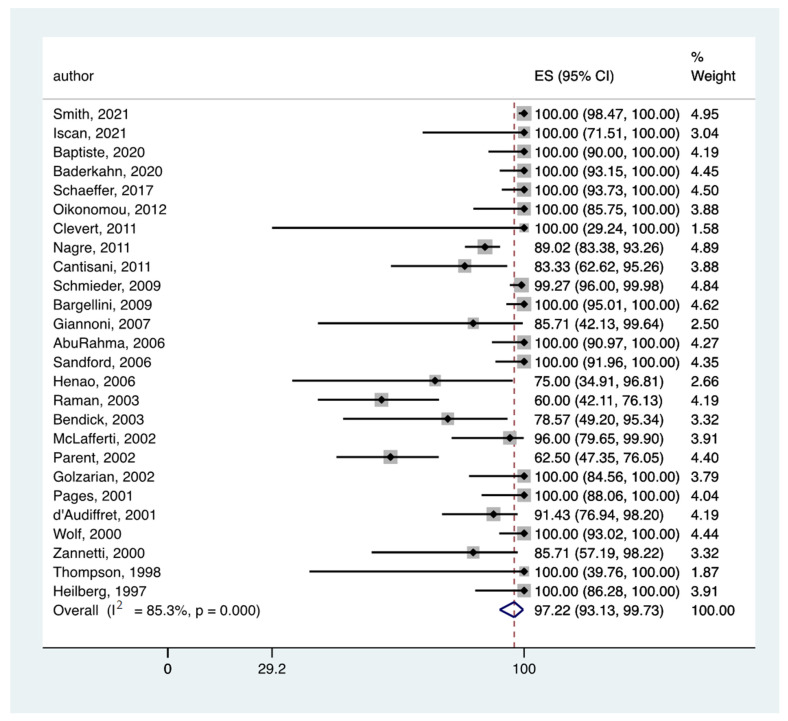
Forest plot presenting the meta-analysis the pooled rate of endoleak detection for CTA. Event rates in the individual studies presented as squares, with 95% confidence interval (CIs) presented as extending lines. The pooled event rate with its 95% CI is depicted as a diamond. ES: Effect Estimate [5,14,15,17,18,19,20,21,22,23,24,25,26,27,28,29,30,31,32,33,34,35,36,37,39].

**Figure 4 jcm-11-03628-f004:**
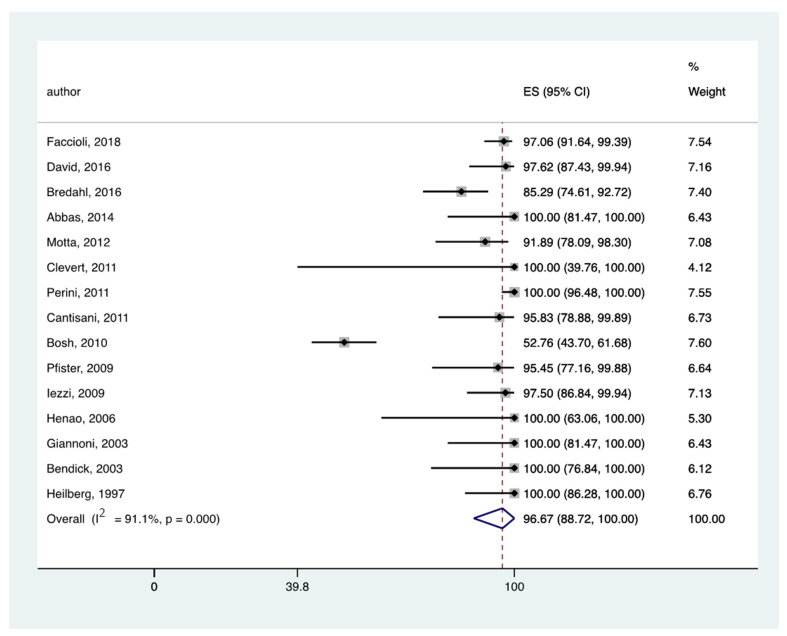
Forest plot presenting the meta-analysis the pooled rate of endoleak detection for CEUS. Event rates in the individual studies presented as squares, with 95% confidence interval (CIs) presented as extending lines. The pooled event rate with its 95% CI is depicted as a diamond. ES: Effect Estimate [9,10,11,19,23,26,28,29,39,40,41,42,43,44].

**Figure 5 jcm-11-03628-f005:**
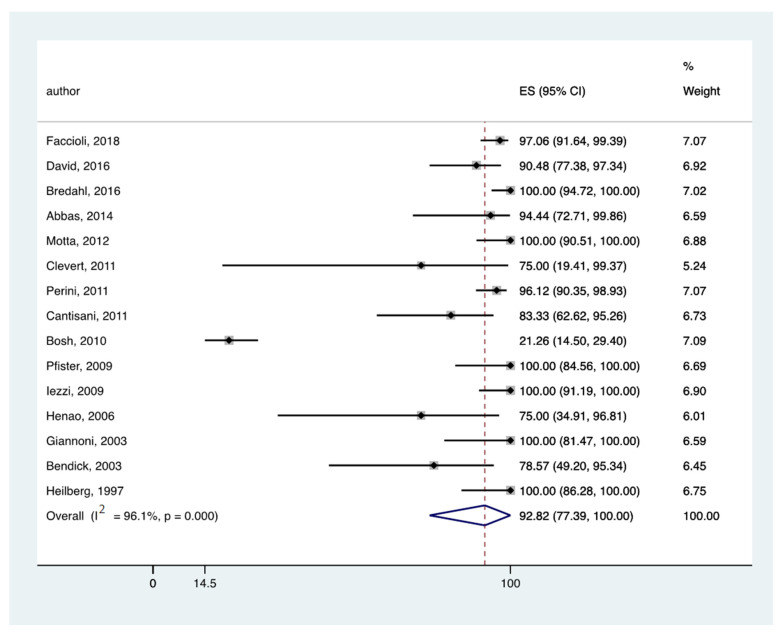
Forest plot presenting the meta-analysis the pooled rate of endoleak detection for CTA. Event rates in the individual studies presented as squares, with 95% confidence interval (CIs) presented as extending lines. The pooled event rate with its 95% CI is depicted as a diamond. ES: Effect Estimate [9,10,11,19,23,26,28,29,39,40,41,42,43,44].

**Table 1 jcm-11-03628-t001:** Baseline study characteristics of included studies comparing CDUS with CTA.

Authors	Country	Year of Publication	Study Design	Period of Recruitment	Number of Participants Who Remained in Surveillance	CDUS Properties for Endoleak Detection (%)					Study Quality NOS
						Sensitivity	Specificity	Accuracy	PPV	NPV	
Smith et al. [5]	London, UK	2021	Comparative Retrospective study	2007–2019	303	NR	NR	NR	NR	NR	8
Iscan et al. [14]	Turkey	2021	Comparative Retrospective study	2018–2020	84	63.6%	100%	95.2%	100%	94.8%	8
Baptiste et al. [15]	France	2020	Comparative Retrospective study	2010–2015	539	58%	92%	NR	39%	81%	7
Baderkahn et al. [17]	Sweden	2020	Comparative Retrospective study	1998–2012	439	88.8%	99.4%	NR	NR	97%	7
Schaeffer et al. [18]	USA	2017	Comparative Retrospective study	2004–2014	88	85%	95%	NR	88%	94%	6
Arsicot et al. [16]	France	2014	Comparative Retrospective study	2010–2012	48	86%	85%	NR	71%	94%	6
Oikonomou et al. [21]	Germany	2012	Comparative Retrospective study	2009–2011	90	75%	95%	NR	85%	91%	5
Clevert et al. [19]	Germany	2011	Comparative Retrospective Study	NR	35	40%	90%	NR	NR	NR	5
Nagre et al. [20]	USA	2011	Comparative Retrospective study	1999–2009	173	NR	NR	NR	NR	NR	5
Cantisani et al. [23]	Italy	2011	Comparative Retrospective study	2007–2009	108	58%	93%	85%	NR	89%	5
Schmieder et al. [24]	USA	2009	Comparative Retrospective study	1996–2007	236	64%	84%	NR	44%	93%	5
Bargellini et al. [22]	Italy	2009	Comparative Retrospective study	1998–2007	184	63%%	98%	93%	85%	93.6%	5
Giannoni et al. [26]	Italy	2007	Comparative Retrospective study	NR	30	NR	NR	NR	NR	NR	5
AbuRahma et al. [25]	USA	2006	Comparative Retrospective study	NR	35	68%	99%	NR	85%	99%	5
Sandford et al. [27]	UK	2006	Comparative Retrospective study	NR	244	64%	91%	NR	52%	95%	5
Henao et al. [29]	USA	2006	Comparative Retrospective Study	2004–2005	20	NR	NR	NR	NR	NR	5
Raman et al. [30]	USA	2003	Comparative Retrospective study	1996–2002	281	43%	96%	NR	54%	94%	5
Bendick et al. [28]	USA	2003	Comparative Retrospective study	NR	40	53%	NR	NR	NR	NR	5
McLafferti et al. [32]	USA	2002	Comparative Retrospective study	1997–1999	79	100%	99%	99%	88%	100%	5
Parent et al. [33]	USA	2002	Comparative Retrospective study	1996–2000	41	NR	NR	NR	NR	NR	5
Golzarian et al. [31]	Belgium	2002	Comparative Retrospective study	1996–1997	55	77%	90%	NR	85%	85%	5
Pages et al. [35]	France	2001	Comparative Retrospective study	1996–1999	41	48%	94%	NR	74%	81%	5
d’Audiffret et al. [34]	France	2001	Comparative Retrospective study	1995–2000	89	96%	94%	NR	89%	98%	5
Wolf et al. [37]	USA	2000	Comparative Retrospective study	1996–1999	76	81%	95%	NR	94%	90%	3
Zannetti et al. [38]	Italy	2000	Comparative Retrospective study	1997–1999	103	66%	100%	82%	100%	71%	3
Thompson et al. [36]	UK	1998	Comparative Retrospective study	1996	20	NR	NR	NR	NR	NR	3
Heilberg et al. [39]	Germany	1997	Comparative Retrospective study	1994–1996	102	NR	NR	NR	NR	NR	3

**NOS:** adjusted Newcastle-Ottawa Scale. Studies could receive a score of minimum 3 to maximum 8 points, **NR:** non reported, **CDUS:** Color-Duplex Ultrasound, **PPV:** Positive Predictive value, **NPV:** Negative Predictive Value.

**Table 2 jcm-11-03628-t002:** Baseline study characteristics of included studies comparing CEUS with CTA.

Authors	Year of Publication	Country	Study Design	Period of Recruitment	Number of Participants who Remained in Surveillance	CEUS Characteristics for Endoleak Detection (%)					Study Quality NOS
						Sensitivity	Specificity	Accuracy	PPV	NPV	
Faccioli et al. [41]	2018	Italy	Comparative Retrospective Study	2011–2016	137	97.1%	100%	98.0%	100%	92.1%	8
David et al. [40]	2016	Italy	Comparative Retrospective Study	2009–2014	181	97.6%	100%	NR	NR	99.3%	7
Bredahl et al. [43]	2016	Denmark	Comparative Retrospective Study	2001–2014	278	85.3%	95%	NR	NR	NR	7
Abbas et al. [42]	2014	Manchester, UK	Comparative Retrospective Study	2012–2013	23	100%	92%	NR	94%	100%	7
Motta et al. [44]	2012	Italy	Comparative Retrospective study	2008–2010	88	92.0%	100%	NR	100%	97.2%	5
Clevert et al. [19]	2011	Germany	Comparative Retrospective Study	NR	35	100%	100%	NR	NR	NR	5
Perini et al. [9]	2011	France	Comparative Retrospective Study	2006–2010	395	NR	NR	NR	NR	NR	5
Cantisani et al. [23]	2011	Italy	Comparative Retrospective study	2007–2009	108	96%	100%	99%	NR	99%	5
Bosh et al. [10]	2010	Netherland	Comparative Retrospective Study	2006–2008	83	NR	NR	NR	NR	NR	5
Pfister et al. [45]	2009	Germany	Comparative Retrospective Study	NR	30	99%	85%	NR	91%	44%	5
Iezzi et al. [11]	2009	Italy	Comparative Retrospective Study	NR	84	97%	82%	89%	NR	97%	5
Henao et al. [29]	2006	USA	Comparative Retrospective Study	2004–2005	20	NR	NR	NR	NR	NR	5
Giannoni et al. [26]	2003	Italy	Comparative Retrospective Study	NR	27	44%	94%	NR	NR	NR	5
Bendick et al. [28]	2003	USA	Comparative Retrospective study	NR	40	93%	NR	NR	NR	NR	5
Heilberg et al. [39]	1997	Germany	Comparative Retrospective study	1994–1996	102	NR	NR	NR	NR	NR	3

**NOS:** adjusted Newcastle-Ottawa Scale. Studies could receive a score of minimum 3 to maximum 8 points, **NR:** non reported, **CEUS:** Contrast enhanced Ultrasound, **PPV:** Positive Predictive value, **NPV:** Negative Predictive Value.

## Data Availability

All data are available in the article. We will willingly share our knowledge, protocol and expertise when asked.

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
