# Peer review of "Colour Duplex and/or Contrast-Enhanced Ultrasound Compared with Computed Tomography Angiography for Endoleak Detection after Endovascular Abdominal Aortic Aneurysm Repair: A Systematic Review and Meta-Analysis"

_jcm, 2022, doi:10.3390/jcm11133628_

Round 1
Reviewer 1 Report
Dear author,
This meta-analysis assess the role of Color Duplex Ultrasound with or without contrast media in EVAR surveillance.
The data exposed in this meta-analysis show the better sensitivity of CEUS over CTA and CDUS, and confim a trend in Vascular Diagnostic, despite its operator-dependency.
As mentioned in this paper limits, it would have been usefull to the absence of a second or third operator repeating the Ultrasound. These data could have been usefull in order to test the real sensitivity with the inter-observer deviation.
Nevertheless, the literature seems to lack in more recent studies about CEUS. In fact the last article about CEUS in EVAR surveillance was published in 2018, with a period of recruitment from 2010 and 2016.
Would have been interesting to understand if the period influence the accuracy, specificity and sensitivity ? Was reported the software and hardware used to perform CDUS? in any case? Were there any significant changes in the software and hardware used?
Author Response
Thank you for this critical issue. We fully agree that DUS and CEUS need to be repeated by a second a probably third operator, as this is an operator-dependent examination. However, not all papers reported inter-observer variability and as a result, we cannot pool this difference in repeated measures, which could have added value to the generalizability of the results. We also agree that the time may have been an important co-variable. However, the time span of six years (2010-2016)is a short period to allow for the detection of major changes in CEUS results among the eligible studies. Moreover, the authors of the studies included in the meta-analysis reported a great variety of hardware and software used for CEUS. As a result, no sound conclusions can be made regarding the effect of equipment upon CEUS results.
Reviewer 2 Report
The stu dy is well conducted and well written.
Author Response
Thank you very much for your interest to evaluate our study
Reviewer 3 Report
This is a systematic review and meta-analysis about detection tool for endoleak after EVAR.
I have several comments.
1. In the snowballing process, how many authors performed this process?
2. line 96. “CEDUS”. When using abbreviation in the first time, please describe full spell.
Author Response
Comment 1: Response
The snowball process has been performed by two independent authors. This statement is reported on page 5, lines 16-17 in the revised manuscript.
"Two authors (GK, CNA) independently extracted and analyzed data and the final decision was reached by consensus"
Comment 2: Response
Thank you very much for the comment. The abbreviation of "CEDUS" was substituted by the abbreviation of "CEUS" and the full spell is reported on page 4 line 17.
"The adjunction of contrast media to the CDUS (Contrast-Enhanced Ultrasound/CEUS)"
Reviewer 4 Report
Title:
In the title of the article, we understand that the authors compare only Colour Duplex with Contrast-enhanced ultrasound for the detection of endoleaks. However, all results were compared with CTA. Therefore, please include the CTA in the title.
Abstract:
Line 34, 35: The percentages are not the same as those obtained from the results. Please correct.
Introduction:
OK
Methods:
“Studies focusing only in one imaging technique were excluded from the analysis.”What was the reason for excluding these studies?
How did you deal with the studies comparing CDUS with CEUS (Study 11)?
The authors use the abbreviation CEUS and sometimes CEDUS lines 94, 96, and 111, for example. Please correct.
Results:
Line 222-223: “Eight studies with total number of 1196 patients comparing CDUS with CTA reported data regarding the comorbidities of the participants.” Why did the authors exclude the CEUS studies from calculating the comorbidities of the participants?
Discussion:
The pooled rate of endoleak detection of CTA was 97.22% when compared to CDUS and 92.82% when compared to CEUS. Can you discuss this difference?
Author Response
Comment 1: Thank you very much for your constructive comment. The title has been accordingly modified in the revised manuscript.
Comment 2:The percentages have been corrected in the Abstract section on page 3, lines 10-11
Comment 3: Thank you very much for the Introduction section
Comment 4 (Methods): Thank you very much for your comments. We performed this meta-analysis to compare CDUS or CEUS with CTA for Endoleak detection after EVAR. As a result, only comparative were available for the present study. This statement is reported in the Inclusion and Exclusion Criteria section, on page 6, lines 1-2.
Comment 5: Thank you very much for this comment. Studies that provided data in both imaging techniques but separately have been included in the present study. Indeed, the study by Iezzi et al. provided data for both imaging techniques compared with CTA in the same manuscript, and they have been extracted and assessed separately.
Comment 6: The abbreviation has been corrected
Comment 7(Results): Thank you very much for the comment. The studies that compared CEUS and CTA did not provide data regarding the comorbidities of the patients.
Comment 8 (Discussion): Great question! Thank you for clarifying this. The discrepancy between the results of CTA endoleak detection obtained from the comparison of CTA vs CDUS and that of CTA vs CEUS is because a different set of studies participated in each comparison. As a result, although there was a common separator, CTA, results were slightly different.
Reviewer 5 Report
I read with great interest your study Colour Duplex and /or Contrast-enhanced ultrasound for endoleak detection after endovascular abdominal aortic aneurysm repair: A systematic review and meta-analysis. I find the analysis well presented and an interesting study topic. The flow chart appears clunky and cumbersome, would not suggest a strictly male pronoun as a universal standard for a bedside imager.
Author Response
Thank you for your supportive comments. We agree that the flow chart may be somewhat not entirely comprehensive. However, this was derived from the PRISMA Guidelines flow Chart Template and we think that we should be in line with the Guidelines document. Of Course, if necessary, please indicate an appropriate flow chart template to reconsider. Moreover, we made some graphical improvements on page 24 of the revised manuscript.